# Ancient DNA Studies in Pre-Columbian Mesoamerica

**DOI:** 10.3390/genes11111346

**Published:** 2020-11-13

**Authors:** Xavier Roca-Rada, Yassine Souilmi, João C. Teixeira, Bastien Llamas

**Affiliations:** 1Australian Centre for Ancient DNA, School of Biological Sciences, University of Adelaide, Adelaide, SA 5005, Australia; yassine.souilmi@adelaide.edu.au (Y.S.); joao.teixeira@adelaide.edu.au (J.C.T.); 2National Centre for Indigenous Genomics, Australian National University, Canberra, ACT 0200, Australia; 3Environment Institute, University of Adelaide, Adelaide, SA 5005, Australia; 4Centre of Excellence for Australian Biodiversity and Heritage (CABAH), School of Biological Sciences, University of Adelaide, Adelaide, SA 5005, Australia

**Keywords:** ancient DNA, Mesoamerica, Teotihuacan, mtDNA, Native American founding lineages, Native American genetic history, Native American ancestries

## Abstract

Mesoamerica is a historically and culturally defined geographic area comprising current central and south Mexico, Belize, Guatemala, El Salvador, and border regions of Honduras, western Nicaragua, and northwestern Costa Rica. The permanent settling of Mesoamerica was accompanied by the development of agriculture and pottery manufacturing (2500 BCE–150 CE), which led to the rise of several cultures connected by commerce and farming. Hence, Mesoamericans probably carried an invaluable genetic diversity partly lost during the Spanish conquest and the subsequent colonial period. Mesoamerican ancient DNA (aDNA) research has mainly focused on the study of mitochondrial DNA in the Basin of Mexico and the Yucatán Peninsula and its nearby territories, particularly during the Postclassic period (900–1519 CE). Despite limitations associated with the poor preservation of samples in tropical areas, recent methodological improvements pave the way for a deeper analysis of Mesoamerica. Here, we review how aDNA research has helped discern population dynamics patterns in the pre-Columbian Mesoamerican context, how it supports archaeological, linguistic, and anthropological conclusions, and finally, how it offers new working hypotheses.

## 1. Introduction

Genomics, namely the study of the complete set of genetic information carried by an individual, has become a powerful tool to map the peopling of the world in detail. Ancient DNA (aDNA) adds a crucial layer of information by providing a snapshot of genetic diversity in past populations, most of which is often hardly accessible in modern-day populations due to recent demographic events. Therefore, aDNA provides evidence in addition to archaeology, linguistics, and anthropology to better comprehend human expansions and demographic events. 

Within the Americas, the genetic history of the Arctic, northern North America, and South America have been recently studied using aDNA [1,2,3,4]. However, aDNA research in populations from southern North America and Central America is extremely limited, despite a rich archaeological record and Indigenous cultural diversity. This is mostly due to the highly degraded nature of genetic material found in archaeological remains [5], further exacerbated by tropical environments or burial practices such as covering the dead with cinnabar. Nevertheless, recent advances in the recovery of authentic aDNA and the application of high-throughput DNA sequencing technologies are enabling the reconstruction of human history in tropical environments, such as the Caribbean islands [6,7,8] and Central America [9], opening the door to a deep exploration of ancient Mesoamerica. 

Mesoamerica is a historically and culturally defined geographic area comprising current central and south Mexico, Belize, Guatemala, El Salvador, and border regions of Honduras, western Nicaragua, and northwestern Costa Rica [10] (Figure 1). Mesoamerica was once home to some of the most famous pre-Columbian American cultures, including the Mayas, Aztecs, and Teotihuacanos. These past cultures developed advanced architecture (step pyramids and ball courts with rings), social organisation, commercial cooperation and trade, culture (hieroglyphic writing and a calendrical system), art (pyrite mirrors and ceramics), politics, an intensive agricultural lifestyle, and in some cases, specific rituals such as human sacrifices [11].

Archaeological evidence indicates a continuous occupation of Mesoamerica since the end of the Late Pleistocene, specifically the Lithic period (~13000–5000 BCE) (Table 1) [11]. However, recent studies provided controversial evidence of human presence in the central-northern region of Mexico, relatively close to Mesoamerica, during the Last Glacial Maximum (26,500–19,000 years ago) [12,13]. Notwithstanding, the first human populations inhabiting this region were nomadic hunter-gatherers. This subsistence strategy lasted until the Archaic (or Proto-Neolithic) period (5000–2500 BCE) when human societies started to transition into semi-sedentary foraging and farming. This lifestyle change was essential for the subsequent development of complex societal structure [11,14], even though the process was not uniform throughout Mesoamerica and differed across time periods and geographical regions. 

The permanent settling of Mesoamerica took place in the Preclassic (or Formative) period (2500 BCE–150 CE), with the development of agriculture and pottery manufacturing. Agricultural practices intensified and improved during the Classic and Epiclassic periods (150–900 CE). The number of large population centres rapidly increased and cultural differences incremented between regions. During the Postclassic period (900–1519 CE), Mesoamerica was inhabited by large intermingled cultural groups connected by trade and agriculture, which fostered migrations that intensified admixture and built up the diversity patterns found across Mesoamerica. The Postclassic period ended with the start of the Spanish conquest (1519–1521 CE) and the subsequent colonial period (1521–1821 CE) [11,15]. The Spanish occupation had extreme cultural and demographic impacts: high mortality rate among Indigenous people, the extinction of languages and cultures, several massive population displacements, and admixture between Indigenous and European, African, and Asian populations [16,17].

## 2. Mesoamerican Genetic Studies 

Mesoamerica was home to different human cultures connected by commerce and farming, and whose varied genetic backgrounds created an invaluable source of genetic diversity in the continent. Previous studies have attempted to uncover the peopling process of Mesoamerica using genetic approaches on modern-day Indigenous and cosmopolitan populations, mostly from Mexico and Guatemala. These analyses mainly focused on studying the so-called uniparental markers: the mitochondrial DNA (mtDNA) [17,18,19,20,21,22,23,24,25,26,27] and the Y-chromosome [17,19,28,29,30], which allow for demographic inference of maternal and paternal ancestries, respectively. Other studies further analysed single-nucleotide polymorphisms (SNPs—variable genomic positions that are present at a frequency ≥1% in the population) located on the autosomes (non-sex chromosomes) [31,32], or sequenced complete exomes (genomic regions that encode proteins) in order to describe Mexican genetic variation, population history, adaptation to the environment, and potential implications for biomedical research [33]. Importantly, this body of work contributes valuable data to describe the global human genetic diversity, especially since Indigenous populations from the Americas are among the most underrepresented populations in human genomic studies [34,35,36].

Most of these genetic studies proposed a single origin for Mesoamerican populations [27,28,29,33], instead of a dual origin as previously argued by Mizuno et al. (2014) [26]. Moreover, Sandoval and colleagues [22,29] analysed the Indigenous population substructure. They demonstrated that the patterns observed by analysing uniparental markers do not display significant population stratification in relation to linguistics, implying that genetic divergence preceded linguistic diversification. It was also shown that modern-day Mesoamerican-related populations contain higher levels of genetic diversity and lower levels of autozygosity (DNA segments identical by descent) compared to modern populations related to other cultural areas, such as Aridoamerica (Figure 1) [25,27,29,32]. These observations agree with the hypothesised importance of agriculture and trade in pre-Columbian Mesoamerica compared to the Aridoamerican hunter-gatherer lifestyle from the northern regions, to foster population movements and admixture that increase genome-wide levels of genetic diversity. 

## 3. Ancient DNA Studies

Inferring the population history of Mesoamerica through genetic studies that solely incorporate modern-day populations can be challenging due to recent historical events, such as population admixture with European, African, or Asian populations, that can distort pre-colonisation genetic signals. For instance, present-day Mexicans have, in general, three different ancestries: Native American, European (closely related to Iberians), and African. While the African component is generally minimal, the European component has a much lower proportion in Native Mexicans than what is found in cosmopolitan Mexicans. Notably, the Native American ancestry component can be divided into six separate ancestries [32]. Three of these are restricted to isolated populations of the northwest, southeast, and southwest, while the other three are widely geographically distributed [32]. Hence, it is essential to include pre-colonisation samples from the archaeological record to contextualise the true genetic diversity and population substructure of ancient Mesoamerica and identify genetic ancestry that might have been lost since the arrival of the Spanish. 

In spite of the extensive archaeological, linguistic, and anthropological research on pre-Columbian Mesoamerica, there have been surprisingly few aDNA studies. Therefore, there is a current lack of genetic knowledge about the genetic history of the region, probably because of the highly degraded nature of genetic material from tropical environments. Thus, currently available ancient genetic data rely mostly on mtDNA studies. The human mtDNA is a circular double-stranded DNA molecule that is maternally inherited, does not recombine [37,38], and has a higher mutation rate compared to the nuclear genome [39]. The sizes of the human reference mtDNA sequences rCRS [40] and RSRS [41] are small at 16,569 base pairs. These characteristics make mtDNA a suitable tool for maternal phylogenetic and phylogeographic inferences [42]. In addition, the high copy number of mtDNA in cells increases the odds of successfully obtaining ancient mtDNA in archaeological remains [43,44]. Mitochondrial haplogroups are lineages defined by shared genetic variation. They have been used to represent the major branches along the mitochondrial phylogenetic tree, unravelling the evolutionary paths of maternal lineages back to human origins in Africa and the subsequent peopling of the world. The Indigenous haplogroups found in present-day non-Arctic populations from the Americas are A2, B2, C1, D1, and D4h3a. All of them descend from a founding population that split from Northern Eurasian lineages ~25,000 years ago and remained stranded in Beringia for ~6000 years before starting to colonise the Americas ~16,000 years ago [45,46,47]. However, there are limitations associated with the study of mtDNA. Indeed, mtDNA represents the evolutionary history of the female population at a single locus and sample sizes are often small. Therefore, it is necessary to remain cautious when interpreting mtDNA results mainly when there is disagreement with those obtained from genome-wide analysis. Nevertheless, most pre-Columbian Mesoamerican genetic data (Table 2 and Table 3) originate from general diagnosis of mtDNA haplogroup variants in the control and/or coding regions. This early technique tested restriction fragment length polymorphisms (RFLPs) which, in this case, result from the presence or absence of mitochondrial sites cleaved by restriction enzymes in order to characterise the mitochondrial haplogroups. Although RFLPs were reported in earlier studies, the partial or complete sequencing of the mitochondrial Hypervariable Regions 1 and 2 (HVR1 and HVR2, respectively) rapidly became the most common strategy to study mtDNA history. Only a few studies in Mesoamerica have been recently able to reconstruct complete ancient mitogenomes [9,48,49] and only one has also generated and analysed genomic data [3].

Interestingly, existing aDNA data come from the Central and Southeast areas of Mesoamerica, corresponding to the Basin of Mexico and the Yucatán Peninsula and its nearby territories, respectively. Therefore, this review will focus on the research that has been performed in these two geographical areas. Nonetheless, it should be noted that Mesoamerica contains other regions with extensive archaeological and anthropological records that have not been investigated in aDNA studies yet. These regions include the Gulf Coast and Oaxaca. The Gulf Coast was initially peopled by the Olmecs (the earliest known major Mesoamerican civilisation) and was later home to the city of El Tajín. Oaxaca comprises the current Mexican state of Oaxaca as well as the border regions of Guerrero, Puebla, and Veracruz. It was home to the Mixtec and Zapotec civilisations, the Preclassic–Classic city of Monte Albán being the most important archaeological site. In contrast, the northern and western regions of Mesoamerica are largely understudied from an archaeological and anthropological perspective and have been severely looted. The North was the transition area between Mesoamerica and Aridoamerica inhabited by both hunter-gatherers and farmers and it was nearly abandoned during the Postclassic period. The West was a geographically diverse area that led to the rise of highly diverse societies characterised by the shaft tomb tradition in the Preclassic and Classic periods and the Tarascan state during the Postclassic [11].

## 4. Basin of Mexico

The Basin of Mexico is a highland plateau that presently includes Mexico City and parts of the states of Mexico, Hidalgo, Tlaxcala, Morelos, and Puebla. The Basin of Mexico previously contained five lakes drained by Spanish colonists in order to expand Mexico City and prevent flooding (Figure 2). This area used to be the homeland of several Mesoamerican civilisations, including the Classic city of Teotihuacan, the Epiclassic city of Cholula, and the Postclassic Toltec, Tepanec, and Aztec Empires [11]. Some aDNA studies have focused on this region, including individuals from the Postclassic period carrying haplogroups A and B at higher frequencies (Table 2) [48,50,51,52,53,54,55,56,57,58,63]. 

### 4.1. Preclassic Period

Mesoamerica’s settling took place during the Early Preclassic period with the emergence of agrarian villages with egalitarian societies. The Middle Preclassic was defined by the development of agriculture, the increase in population sizes in settlements with hierarchical societies, and a strong influence from the Gulf Coast. This influence later disappeared during the Late Preclassic, when some settlements (e.g., Cuicuilco) became capital towns with strong economic and political power. At the end of this period, Teotihuacan rose as the undisputed power in Mesoamerica [11]. 

### 4.2. Classic Period

Teotihuacan was a powerful political and military centre that controlled trade routes and attracted large masses of people from distant regions, coordinating them into an effective multi-ethnic system. The metropolis grew into the largest city of Mesoamerica and one of the world’s largest and most urbanised population centres of the first millennium, reaching 80,000–100,000 inhabitants by approximately 150 CE [64]. Teotihuacan’s art and architecture styles are widespread in Mesoamerica, suggesting that the city had extensive influence and acted as the principal trade metropolis in the region. Obsidian tools were the main export (along with ceramics and other luxury goods) due to the presence close to the city of the most important deposit of obsidian in Mesoamerica [11].

Teotihuacan was first occupied ~500 BCE and consisted of villages of local people as the valley offered several advantages: an abundance of freshwater springs, volcanic and raw materials for construction and technology, and a crucial location along the route from the Gulf Coast to the Basin of Mexico, skirting the Trans-Mexican Volcanic Belt also known as Sierra Nevada [64].

During the first century CE, Teotihuacan was the destination of numerous migrations from Puebla and Tlaxcala, probably due to the Popocatépetl volcano eruption [65]. It has been proposed that the metropolis was either controlled by powerful rulers and military elites [66,67] or coordinated as a corporate co-rulership system, where each one of the four seats of the state was occupied by one of the four administrative sectors of the city [64]. 

Around 350 CE, a new episode of population displacement from Cuicuilco and surrounding regions likely took place when the Xitle volcano erupted [68]. This migration coupled with a possible rivalry between two of the seats of the state may have generated a political crisis that climaxed with the destruction and burning of the Temple of the Feathered Serpent [64] located in the Avenue of the Dead, the city’s main avenue. The Moon Pyramid is located on the same avenue and underwent a reconstruction approximately during this period. Interestingly, ancient DNA was extracted from a human archaeological sample excavated from the Moon Pyramid. Because of poor DNA preservation, and despite using advanced high-throughput sequencing techniques, investigators had to rely on imputation methods to assign the mitochondrial subhaplogroup A2 to the individual. A major limitation of this study is the unavailability of a panel from a population both closely related to and contemporaneous with the Moon Pyramid individual to perm imputations with the highest level of confidence [63]. 

Following the political crisis, Teotihuacan became a multi-ethnic city inhabited by locals, returnee migrants, people from the Tlaxcala–Hidalgo–Puebla corridor, and individuals from distant regions, such as the Gulf Coast, Oaxaca, and Chiapas. This pattern has been sustained by archaeological studies [64], trace element analysis [69], stable isotopes [70], and strontium isotopes [71]. The city’s internal structure consisted of various neighbourhoods, each one specialised in producing specific goods and supervised by the intermediate elites whose power rapidly increased, changing Teotihuacan’s political organisation [64]. On the one hand, most neighbourhoods were segregated by the province of origin of its inhabitants, such as Tlailotlacan, home to migrants with an Oaxaca origin [64]. Genetic analyses on eight individuals from Tlailotlacan revealed only the mtDNA haplogroups A and D [50]. On the other hand, some neighbourhoods were multi-ethnic settlements. Aguirre-Samudio and colleagues [51] examined 36 individuals from three different multi-ethnic neighbourhoods (San Sebastián Xolalpan, La Ventilla, and San Francisco Mazapa). They found that the most frequent mtDNA haplogroup was A (58%), followed by haplogroups B (25%), C (14%), and D (3%). Furthermore, the genetic diversity of the analysed mitochondrial sequences was higher compared to the Tlailotlacan individuals, which supports the multiethnicity hypothesis. Additionally, two studies by Álvarez-Sandoval and colleagues [52,53] also supported this hypothesis using 29 individuals from Teopancazco, another multi-ethnic neighbourhood [64]. They also found the four autochthonous mtDNA haplogroups (A, B, C, and D), but reported no statistically significant differences between haplogroup frequencies across periods. Although the levels of genetic diversity between periods were not significant either, they suggested that the population from Teopancazco was heterogeneous since the beginning of its settlement and that diversity increased over time [52,53,64]. 

Around 550 CE, Teotihuacan’s population size declined by half, and the city collapsed without any clear explanation. Although some of the earliest theories postulated that Teotihuacan was sacked by outside invaders [66], the most recent archaeological studies have proposed alternative explanations. Manzanilla [64] suggests that disagreements between the corporate structure of the state and the exclusionary structure of the strong intermediate elites eventually reached a breaking point that tore the city’s social and political organisation apart. It has also been proposed that Teotihuacan’s influence in Mesoamerica began to weaken, as many of the conquered cities became increasingly autonomous, which could have caused an economic crisis and led to a revolt [72]. The archaeological evidence for both of these theories is the deliberate burning of major buildings and the destruction of artworks and religious sculptures [64]. However, the specific destruction of effigies of Tlaloc (the God of the Rain) [73,74] has been argued to instead represent the abandonment feeling that Teotihuacanos might have had towards their God. It has been suggested that during the end of the Classic period, a severe drought caused population loss and abandonment of large urban centres and villages throughout all Mesoamerica [11,74,75]. Teotihuacan utilised freshwater springs for irrigation and domestic consumption, whereby a severe drought would have affected the agriculturalist lifestyle, preventing the growth of staple food crops, such as maize, beans, avocados, peppers, and squash, subsequently affecting the breeding of farm animals such as chickens and turkeys [64,69]. This situation might have caused famine and probably the spread of epidemic diseases, such as haemorrhagic fevers, as hypothesised by Acuna-Soto and colleagues [75] after demonstrating a synchronicity of climatological and demographic events during the Classic period and the 16th century in Mesoamerica [76]. Lastly, it has also been proposed that the agriculturalist lifestyle was instead affected by a possible depletion of cultivated soils and the overexploitation of nearby wooded areas [11]. Insofar, the fall of Teotihuacan and the collapse of the Classic period in Mesoamerica remain a mystery.

### 4.3. Epiclassic Period

The Epiclassic, also known as Late Classic in other parts of Mesoamerica, was marked by the decadence of Teotihuacan and extensive population movements. Various ephemeral centres of power emerged throughout the Basin of Mexico, such as Xochicalco, Teotenango, Cacaxtla, and Cholula [11]. The latter was a sacred city in the Puebla–Tlaxcala Valley that had been continuously occupied since ~100 BCE and was heavily affected by the Spanish conquest and colonisation. After Teotihuacan’s collapse, the population is thought to have migrated to Cholula, with which they were culturally associated since the Classic period [77]. Even though Cholula gained power in the Basin of Mexico during the Epiclassic period, it was highly influenced by the Gulf Coast culture until the Postclassic period [11]. Morales-Arce and colleagues (2019) [48] analysed 12 individuals from Cholula, dating from the Classic to the Middle Postclassic to assess the population dynamics around this time. The study determined that haplogroups A and B were equally the most frequent (42%), haplogroup C was the rarest (16%), and haplogroup D was absent. Moreover, a previous study by Juárez Martín [58] analysed nine individuals from the Postclassic that all carried haplogroup A. Interestingly, the genetic distance between the Cholula [48] and Teopancazco individuals [53] was higher than any other Central Mesoamerica population, which challenges the hypothesised migration from Teotihuacan to Cholula based on the archaeological record.

### 4.4. Postclassic Period

The study of the Postclassic includes not only archaeology and anthropology, but also linguistic information from accounts written in Spanish and different Indigenous languages. While the Classic period was a peaceful epoch of cultural climax, the Postclassic period was marked by political instability and war. This period emerges with the apogee of Tula (also known as Tollan-Xicocotitlan), the capital of the Toltec Empire. Even though the political system and area of influence of the Toltec culture remain a point of discussion among scholars, especially in the city of Chichen Itza, inhabited by the Mayas in the Yucatán Peninsula, it has been proposed that its influence was widespread and persisted through time in central Mesoamerica [11]. 

After the collapse of the Toltec Empire, numerous city-states emerged and fought one another for the control of political power. Even though Azcapotzalco (capital of the Tepanec state) initially ruled most of the Basin of Mexico, the Triple Alliance of the city-states of Tenochtitlan, Texcoco, and Tlacopan reduced Tepanec influence and allowed the rise of the Aztec Empire. The Aztecs conquered a large number of cities across Mesoamerica over the following years and established a large Empire that lasted until the Spanish conquest [78]. For example, the city of Xaltocan, founded by Otomi-speaking people around 1100 CE, was initially conquered by the Tepanec state and later integrated into the Aztec Empire [79]. Colonial documents clearly state that the Tepanec conquest ended with the fleeing of the original Otomi population and that the Aztec conquest led to population replacements from various places in central Mesoamerica and the Gulf Coast to Xaltocan [80]. Nevertheless, archaeological evidence suggests a population continuity after the first defeat and even during the incorporation into the Aztec empire [81]. A single study has so far focused on the genetic impact of Aztec Imperialism in the archaeological site of Xaltocan. Mata-Míguez and colleagues [54] analysed mtDNA haplogroups in 10 pre- and 15 post-Aztec conquest individuals and found that haplogroups A2 and C1 increased over time (from 30% to 60% and from 0% to ~7%, respectively), whereas haplogroups B2 and D1 decreased over time (from 30% to 20% and from 40% to ~13%, respectively). The study evidenced that some haplotypes were found in multiple individuals from the same time period, indicating close maternal relationships. However, the distinct haplotypes found across time periods are unlikely to have resulted from genetic drift or the occurrence of new mutations, given the short evolutionary time across samples. Instead, results are more consistent with a replacement of maternal lines in the sampled Xaltocan individuals after the Aztec conquest.

In contrast, various studies have focused on the archaeological site of Tlatelolco, which is buried beneath modern-day Mexico City and was founded around 1337 CE. Even though Tlatelolco was a sister city to Tenochtitlan (the capital of the Aztec Empire), religious and cultural practices were distinct between the two cities. Tlatelolco was the biggest and most diversified marketplace in Mesoamerica during the Late Postclassic [11]. The most recent aDNA study was performed by Morales-Arce et al. (2019) [48] on 11 subadults found in a sacrificial ritual context. Results show that 55% of the individuals harboured the haplogroup A, 18% both the haplogroups B and D, and 9% the haplogroup C. Similar frequencies were reported in three previous studies conducted by Kemp et al. (2005) [55] with 23 adults from a burial at the ceremonial centre of Tlatelolco, Solorzano Navarro (2006) [56] with 30 individuals from the same burial site, and De La Cruz et al. (2008) [57] with 14 subadults from a sacrificial context dedicated to Tlaloc, the God of the Rain.

## 5. Yucatán Peninsula

The Yucatán Peninsula is located in southeastern Mexico and separates the Caribbean Sea from the Gulf of Mexico. Currently, it comprises the Mexican states of Yucatán, Campeche, and Quintana Roo, as well as part of Guatemala and almost the whole of Belize. The first humans inhabiting this area were hunter-gatherers. In fact, the oldest individual (~10500 BCE) with published genetic data in Mesoamerica comes from this region. This individual was found in Hoyo Negro (Figure 3 and Table 3), a submerged collapsed chamber in the Sac Actun cave system, and harboured the autochthonous mitochondrial haplogroup D1 [59,82]. However, this result has been highly debated as contamination was observed and post-mortem damage patterns (a characteristic property of aDNA) were not evident in the mitochondrial sequences belonging to the haplogroup D1 [83].

Haplogroup D1 was actually found in two genetically related individuals (~5300 BCE) from the archaeological site of Saki Tzul in the Maya Mountains (Belize) [3]. Another individual (~7300 BCE) from the same region, found in the archaeological site of Mayahak Cab Pek (Belize), carried the native haplogroup D4h3a [3]. The genomic data of these individuals from the Maya Mountains revealed that they are genetically closer to present-day groups elsewhere in Central and South America rather than from the region near Belize [3]. This result suggests that they formed part of a homogenous population associated with the Clovis culture [84] that expanded southwards carrying the primary source of the ancestry of Central and South Americans.

## 6. Maya Civilisation

The entire Yucatán Peninsula together with the Mexican states of Tabasco and Chiapas, the rest of Guatemala and Belize, as well as the border regions of Honduras and El Salvador, were the home to the Maya civilisation (Figure 3), one of the most advanced and highly developed societies in ancient Mesoamerica [11]. Most aDNA studies performed in this region include individuals from the Postclassic period and the mtDNA haplogroup A is the most frequent (Table 3).

The geographical diversity of the Yucatán Peninsula was one of the most important traits for the Mayan settling in the Preclassic period, allowing the emergence of different cultural focuses that led to the rise of various linguistic families. Despite the existence of a cultural influence from the Gulf Coast during the Middle Preclassic, it disappeared in the Late Preclassic when cities started to urbanise with architectural gigantism [11]. The Classic period in the Maya civilization was defined by the development of the Mayan tradition and the influence from Teotihuacan. For instance, archaeological research has demonstrated how Teotihuacan decisively intervened in Tikal (Guatemala), imposing an ally government [85]. Tikal and Copán (Honduras) were two of the most powerful cities in the Mayan Classic period. During the Late Classic (equivalent to the Epiclassic in the Basin of Mexico), several Mayan cities, especially in the south, collapsed after the fall of Teotihuacan. Despite this circumstance, Teotihuacan’s culture lingered in Mayan civilisation. The Classic collapse caused various population displacements that led to huge demographic events in other cities, most of them in the north, such as Chichen Itza. This population relocation enabled the splendour of the Mayan economy, politics, and culture. The apogee of the northern Mayas climaxed in the Early Postclassic. However, this period was shaped by the arrival of invaders, probably from the Toltec Empire. In the northern Mayan area, the Late Postclassic was defined by the fall of Chichen Itza and the rise of Mayapan, whereas in the southern Mayan area, the Late Postclassic was characterised by the brief power of the K’iche’ (Quiché), Q’umarcaaj being the most famous capital [11]. This period ended with the Spanish conquest and the onset of colonialism. 

Regarding genetic studies of the region, Ochoa-Lugo and colleagues [60] performed a transitional aDNA study analysing 38 Mayan individuals from the Classic until the Postclassic periods across different archaeological sites in Mexico. Haplogroups A and C were the most frequent (61% and 34%, respectively), while haplogroup D was the least frequent (5%). Importantly, the complete absence of haplogroup B in this study, along with its low frequency amongst contemporary Mayan populations found in previous work, seemingly supports the hypothesis that haplogroup B independently entered the Americas through a later, separate migration into the continent [18,60,86,87]. However, this is disputed by genetic evidence showing that all Indigenous mtDNA haplogroups found in present-day Mesoamerica descend from a single founding population [45,46,47].

Additionally, other aDNA studies just focused on the analysis of mtDNA haplogroups in one archaeological site during a specific period of time. For example, the largest mtDNA collection from Mesoamerica to date comes from the Classic–Epiclassic site of Midnight Terror Cave (Belize), which is believed to have been a site of human sacrifice [49]. A total of 17 complete mitogenomes were reconstructed, including haplogroups A2 (82%), B2 (12%), C1 (6%), but not D. Moreover, a total of 24 individuals were genetically sex assigned and more than 60% were female, which raises questions about the hypothesis that young males often captured in battle by the Mayans were the most common sacrificial victims. Nonetheless, De La Cruz and colleagues [57] analysed 14 subadults in an Aztec sacrificial context in the Postclassic Tlatelolco (Basin of Mexico) and molecular sex identification revealed that among nine individuals analysed, eight were male and one was a female. 

Finally, a further study focused on the Epiclassic–Postclassic site of Xcaret (Mexico), one of the most active ports in the eastern Mayan coast [61]. This research included 24 individuals, most of which carried haplogroup A (88%). While haplogroups B and C were found at low frequencies, no haplogroup D was found. These individuals were more related to modern-day Mayans and other contemporary populations of Mesoamerican origin than to the nine Epiclassic–Postclassic individuals from Copán (Honduras), from whom only haplogroups C and D have been reported [62]. 

## 7. Greater Nicoya

The Greater Nicoya was an ancient Central American cultural area currently located in the Pacific Nicaragua and northwestern Costa Rica. It was situated between southern Mesoamerica and northern Isthmo-Colombia (Figure 3). This region was first occupied by Chibchan-speaking groups that shared a cultural and linguistic background with ancient and current populations from Isthmo-Colombia [88]. However, it has been postulated that Mesoamerican populations from the Basin of Mexico migrated southwards and settled in this region in the Epiclassic period (around 800 CE), displacing the previous Isthmo-Colombian populations [11]. The Greater Nicoyan culture climaxed in the Postclassic period and received a strong influence from Mesoamerica, evidenced in the archaeological record through ceramic iconography and religious iconology [88]. At the time of the Spanish arrival, it was documented that the populations from the Greater Nicoya spoke Mesoamerican-related languages [89]. In particular, Morales-Arce and colleagues [9] reconstructed three complete ancient mitogenomes from the Epiclassic-Postclassic archaeological site of Jícaro (Figure 3 and Table 3). Although all of them belonged to the same mitochondrial subhaplogroup (B2d), each individual had a distinct sequence, indicating these individuals were not maternally related in the recent past. The study also suggested that these individuals had a greater degree of maternal genetic affinity with contemporary Isthmo-Colombians, than with contemporary Mesoamericans. 

## 8. Future Perspectives

Recent improvements in the field of aDNA have revolutionised the study of the history of past populations, providing additional information highly valuable for archaeologists, linguists, and anthropologists. Given that Mesoamerica is the geographical link between the North and South American continents, aDNA studies from this region may be essential to understand the migration patterns of populations across the region, as well as their territorial expansions and interactions. For example, the aDNA study performed by Moreno-Mayar and colleagues [2] described a putative northward expansion out of Mesoamerica around 8700 years ago and a second one southward that contributed to the ancestry of most South American groups, except Patagonians. However, this study only included archaeological remains from northern North America and South America, and no Mesoamerican records were considered. Moreover, a recent study proposed a putative contact between Central Americans and Oceanian populations (~1200 CE) using modern DNA and some ancient individuals across the Americas (including the Saki Tzul individuals from the Maya Mountains) [90]. Integrating additional ancient genomes through time and space in Mesoamerica could help to clarify these hypotheses.

Archaeological evidence strongly suggests that after the settling of Mesoamerica, most cultures remained connected by trade, agriculture, culture, politics, empires, and conquests. Therefore, it is essential to include genetic analysis to examine the patterns of population structure, biological adaptation, cultural impacts, and natural phenomena (e.g., volcanos) on the evolutionary and demographic history of Mesoamerican populations. Crucially, aDNA can help to discern these patterns in pre-Columbian Mesoamerica and provide further support to archaeological studies as well as offer new insights and working hypotheses. This is particularly important for the highly understudied Preclassic period, which could benefit from interdisciplinary approaches that help to unravel the demographic and genetic architecture of the settling process of Mesoamerica. 

Regarding the Basin of Mexico, a detailed study focused on samples from the Classic city of Teotihuacan, concretely in multi-ethnic neighbourhoods such as Teopancazco, could not only help to determine aspects of population demography of Teotihuacan, but also be useful to infer and contextualize the population history of Mesoamerica during the Classic period. This should be aided by similar studies focusing on the understudied Epiclassic and the Postclassic periods, which could help to explain how people were connected after the fall of Teotihuacan and during the empires that emerged before the Spanish conquest, respectively. 

Although Mayan culture remained in the Yucatán peninsula throughout all these cultural changes, it was never isolated and was also affected by the Classic collapse. A genetic study focusing on the Mayas could further expand our knowledge of this important civilisation and reveal connections with other regions of Mesoamerica. In fact, a knowledge gap currently exists in several Mesoamerican areas that could help to define population interactions in the region. These include the Gulf Coast, the North, the West, and Oaxaca regions, as well as the cultural and historical areas of Isthmo-Colombia, Aridoamerica, and Oasisamerica.

## 9. Conclusions

Mesoamerica represents a critical tropical transition in the southward journey of the First Peoples of the Americas. A thorough description of genetic variation in ancient Mesoamerican populations through space and time may enable us to identify mechanisms of adaptation to changing environments, climates, lifestyles, and social structures, with potential consequence for present-day people’s health. For instance, this kind of research could target the Mesoamerican Classic collapse and investigate a possible genetic link to severe droughts or epidemic diseases, and potential consequences to the genetic make-up of contemporary Mesoamerican populations. Thus, it is highly important to reconstruct the genetic past of Mesoamerica before and after European arrival and estimate the genetic impact of European colonisation in the demographic and adaptive history of these populations, which currently remains unknown. 

## Figures and Tables

**Figure 1 genes-11-01346-f001:**
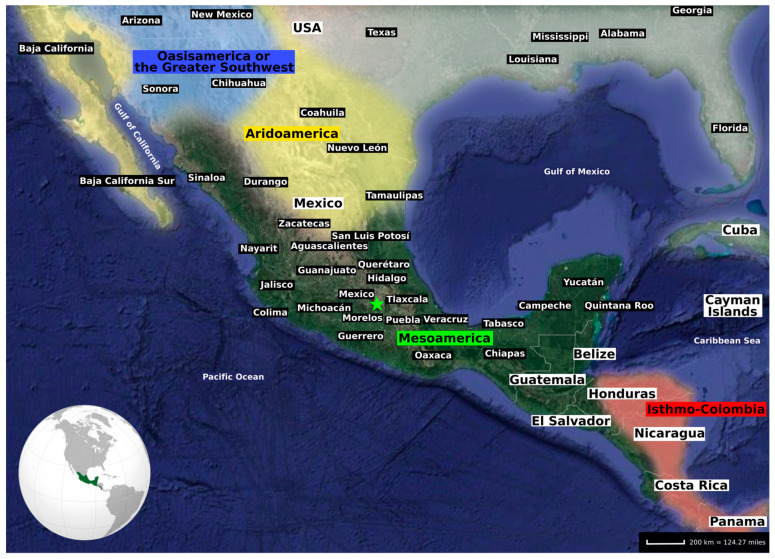
The historical and cultural area of Mesoamerica, as well as Isthmo-Colombia, Aridoamerica, and Oasisamerica (also known as the Greater Southwest) represented with coloured shades. Regions outside these areas are depicted with a white shade. Of note, Oasisamerica is occasionally included in Aridoamerica in the literature. White lines delineate the modern-day borders between countries and white dashed lines the borders between states. The green star is the localisation of present-day Mexico City.

**Figure 2 genes-11-01346-f002:**
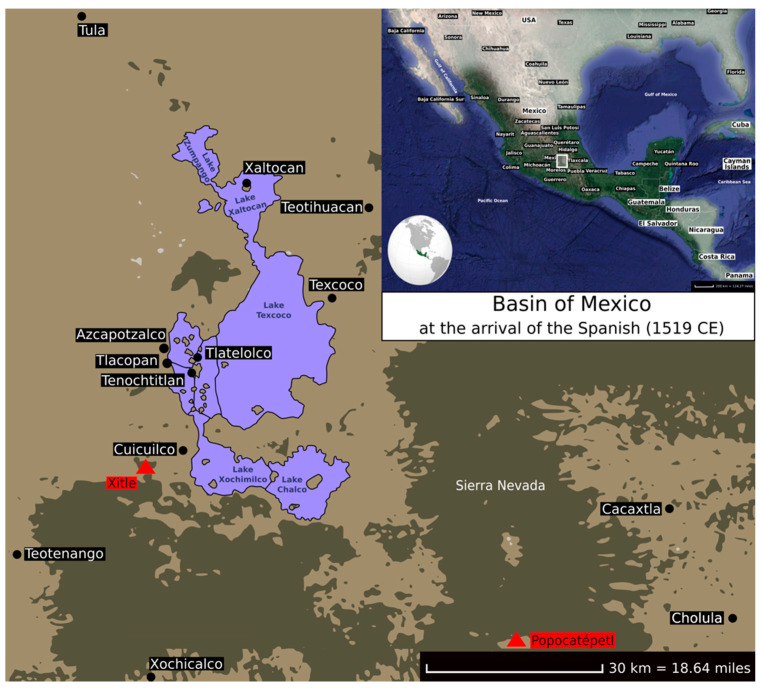
Approximate locations of the archaeological sites (black dots) and volcanoes (red triangles) in the Basin of Mexico at the arrival of the Spanish (1519 CE), as discussed in the present review.

**Figure 3 genes-11-01346-f003:**
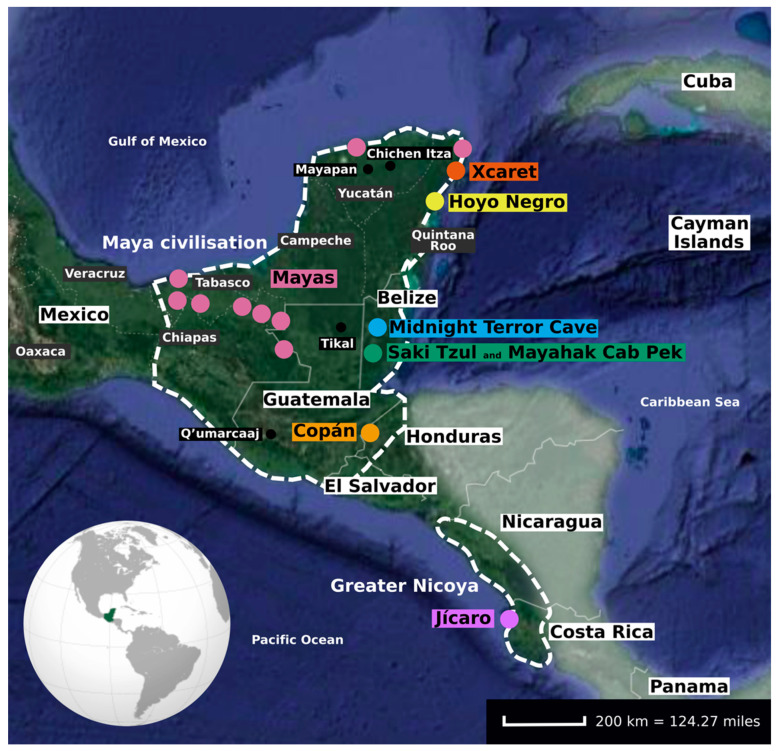
Locations of the archaeological sites in the Yucatán Peninsula and nearby territories, as discussed in the present review. Coloured dots represent the archaeological sites included in each aDNA study. Black dots represent archaeological sites mentioned in the present review with no aDNA data so far. The historical and cultural areas of the Maya civilisation and the Greater Nicoya are within the thick white dashed lines. The shaded area represents regions outside of Mesoamerica. White lines delineate current borders between countries, and thin white dashed lines borders between states.

**Table 1 genes-11-01346-t001:** Mesoamerican chronology: BCE (Before Common Era); CE (Common Era).

Period	Calendar Dates
Lithic	~13000–5000 BCE
Archaic or Proto-Neolithic	5000–2500 BCE
Preclassic or Formative	2500 BCE–150 CE
Classic	150–650 CE
Epiclassic *	650–900 CE
Postclassic	900–1519 ^†^ CE
Colonial	1519 ^†^–1821 ^‡^ CE
Present	1821 ^‡^ CE–Current Day

* The Epiclassic is also known as Late Classic in parts of Mesoamerica. ^†^ Start of the Spanish conquest of Mexico. ^‡^ Mexican Independence Date.

**Table 2 genes-11-01346-t002:** Ancient DNA studies in the Basin of Mexico.

Site	*n*	mtDNA Sequence	Genetic Sex Assign	Nuclear Data	mtDNA Haplogroup Frequency (%)				Date	Period	Location	Reference
					A	B	C	D				
Teotihuacan (Moon Pyramid)	1	Imputed WMG	No	No	100	0	0	0	~500 CE	Classic	State of Mexico, Mexico	Mizuno et al. (2014) [26]
Teotihuacan (Tlailotlacan)	8	GD and HVRI	No	No	50	0	0	50	300–500 CE	Classic		Herrera Salazar (2007) [50]
Teotihuacan (La Ventilla, San Sebastián Xolalpan and San Francisco Mazapa)	36	GD	No	No	58	25	14	3	300–700 CE	Classic–Epiclassic		Aguirre-Samudio et al. (2017) [51]
Teotihuacan (Teopancazco)	29	HVR1 and HRM	Yes	No	55	21	17	7	200–550 CE	Classic		Álvarez-Sandoval et al. (2014) and (2015) [52,53]
Xaltocan (Pre-Aztec conquest)	10	GD and HVR1	No	No	30	30	0	40	1240–1541 CE	Postclassic		Mata-Míguez et al. (2012) [54]
Xaltocan (Post-Aztec conquest)	15				60	20	7	13				
Tlatelolco	23	GD	No	No	65	13	4	18	1325 CE	Postclassic	Mexico City, Mexico	Kemp et al. (2005) [55]
	30	GD and HVR1	No	No	46	37	7	10	1350–1457 CE			Solorzano Navarro (2006) [56]
	14	GD	Yes	No	57	21	7	14	1454–1457 CE			De La Cruz et al. (2008) [57]
	11	HVR1, HVR2, and WMG *	Yes	Yes *	55	18	9	18	1350–1519 CE			Morales-Arce et al. (2019) [48]
Cholula	12				42	42	16	0	240–1400 CE	Classic–Postclassic	State of Puebla, Mexico	
	9	GD and HVRI	No	No	100	0	0	0	1100–1500 CE	Postclassic		Juárez Martín (2002) [58]

HVR1—Hypervariable Region 1 (or a segment); HVR2—Hypervariable Region 2 (or a segment); GD—general diagnostics of mtDNA haplogroup variants in the control and/or coding regions (restriction fragment length polymorphism (RFLP) analyses, sometimes followed by Sanger sequencing); HRM—haplogroup characterisation by high resolution melting analysis; WMG—whole mitogenome. * Some individuals.

**Table 3 genes-11-01346-t003:** Ancient DNA studies in the Yucatán Peninsula, Maya Civilisation, and Greater Nicoya.

Region	Population/ Site	*n*	mtDNA Sequence	Genetic Sex Assign	Nuclear Data	mtDNA Haplogroup Frequency (%)				Date	Period	Location	Reference
						A	B	C	D				
Yucatán Peninsula	Hoyo Negro	1	GD	No	No	0	0	0	100	~10500 BCE	Lithic	State of Quintana Roo, Mexico	Chatters et al. (2014) [59]
	Saki Tzul	2	WMG	Yes	Yes	0	0	0	100	~5300 BCE		Toledo District, Belize	Posth et al. (2018) [3]
	Mayahak Cab Pek	1				0	0	0	100	~7300 BCE			
Maya Civilisation	Mayas	38	HVR1	No	No	61	0	34	5	250–1500 CE	Classic– Postclassic	States of Quintana Roo, Yucatán, Chiapas and Tabasco, Mexico	Ochoa-Lugo et al. (2016) [60]
	Midnight Terror Cave	17	WMG	Yes	No	82	12	6	0	550–900 CE	Classic– Epiclassic *	Cayo District, Belize	Verdugo et al. (2020) [49]
	Xcaret	24	GD	No	No	88	4	8	0	600–1521 CE	Epiclassic *–Postclassic	State of Quintana Roo, Mexico	González-Oliver et al. (2001) [61]
	Copán	9	GD	No	No	0	0	88	12	700–1300 CE		Copán Department, Honduras	Merriwether, Reed, and Ferrell (1997) [62]
Greater Nicoya	Jícaro	3	WMG	No	No	0	100	0	0	800–1250 CE	Epiclassic *–Postclassic	Guanacaste Province, Costa Rica	Morales-Arce et al. (2017) [9]

HVR1—Hypervariable Region 1 (or a segment); GD—general diagnostics of mtDNA haplogroup variants in the control and/or coding regions (restriction fragment length polymorphism (RFLP) analyses, sometimes followed by Sanger sequencing); WMG—whole mitogenome. * The Epiclassic is also known as Late Classic in parts of Mesoamerica.

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
