# Peer review of "Ancient DNA Studies in Pre-Columbian Mesoamerica"

_genes, 2020, doi:10.3390/genes11111346_

Round 1

Reviewer 1 Report

The review entitled "ANCIENT DNA STUDIES IN PRE-COLUMBIAN MESOAMERICA" presents the studies on ancient DNA carried out so far in Mesoamerica, a key region in the settlement of the Americas. 

The Historical explanation to put into context the importance of ancient DNA studies in that region is well done and the works are well presented. However, I suggest that some aspects could be clarified.

General comments:

  • The abstract focuses on the importance of this type of data in this region, but without mentioning some key aspects that would help the reader to get at least a minimum idea of the state of the research. So, I think that the Abstract would benefit if there were mentioned if the data available are mainly from mtDNA or other data exist. Which are the geographic areas and the periods best studied so far, and which have no or few data. For example, it would be good to mention that the two regions with genetic data are the Basin of Mexico and the Yucatan peninsula.

  • After introducing the region and genetic studies, the review focuses directly on the Basin of Mexico, then on the Yucatán Peninsula, and finally the Greater Nicoya. Thus, as a non-specialist in this particular region, I wonder whether apart from these main known cultural centres there are interesting areas of Mesoamerica that have not yet been investigated that would also be interesting. Are there other minor regions, which already remains so far understudied that would be also worth studying. Or the interest of these three regions is so huge that any other remains from the pacific coast for example, or from North of Mexico, is totally secondary? Or whether in fact there are no remains that could be studied from those other regions?. So, I would suggest before focusing on these 3 geographic areas, to mention the reasons why the next three sections are focused on those three particular geographic regions within the context of all Mesoamerica.
  • More info about the only individual with autosomal data would be very interesting, mainly, because all previous studies on aDNA and conclusions are based on mtDNA haplogroup frequencies considering few individuals that may not represent the ancient population. Thus, having autosomal data would be highly valuable.

Minor comments:

  • Line 15: (include date of 1rt migration wave/settlement), otherwise it seems that is the same date than the next sentence (2500BC)
  • Line 37: I am confused with the geographic reference. To what region do the authors refer when mentioning “northern North America” and “southern North America”?. Because in Scheib et al., 2018, the samples they sequence are from North America, but I would not consider that region northern North America. Do they mean Pacific and Atlantic Coast? I see that Moreno-Mayar used these terms (AB, NNA and SNA), but to mention dispersal routes. I suggest to better clarify these geographic regions.
  • Line 86. I don’t understand what authors mean by: “and whose varied ancestry represented a unique source of genetic diversity in the continent.”
  • Line 114: In my opinion the components of Native Mexicans should be better specified. In fact, the African component is minimal, and the European component has a much lower proportion than that found in cosmopolitan Mexicans. The differences in these three components should be stated or otherwise do not refer to Native Mexicans, but Mexicans in general.
  • Line 146: you mention a work (ref 3) that analysed genomic data in Mesoamerica, but this reference earlier in the introduction was cited when talking on northern North America.
  • Figure 2. What do the red triangles mean?
  • Figure 3. What the different colors of the names of the sites represent?  Which are the difference between colored points with the black points?

Typos:

  • Line 67: humans societies --> human societies
  • Line 156: Focused in --> Focused on

Author Response

We would like to thank you for the constructive comments and valuable insights provided, which have contributed to strengthen this manuscript. We thoroughly edited the manuscript to correct spelling mistakes and to improve the readability of some sentences. We also addressed all of the points that you raised as detailed below.

I think that the Abstract would benefit if there were mentioned if the data available are mainly from mtDNA or other data exist. Which are the geographic areas and the periods best studied so far, and which have no or few data. 

We have heavily edited the abstract and we incorporated a comment about how most of the ancient DNA data available in Mesoamerica focus on mtDNA from the Basin of Mexico, the Yucatán Peninsula and its nearby territories, being the Postclassic the most studied period (see line 18). Due to word limit constraints, the sentence: “Archaeological evidence indicates continuous human occupation since shortly after the first migration wave in the Americas” has now been deleted. This is also related to the fact that we want to focus on the time since the settling of Mesoamerica. Therefore, we have not included the date of the first migration wave as per your minor comment (line 15).

After introducing the region and genetic studies, the review focuses directly on the Basin of Mexico, then on the Yucatán Peninsula, and finally the Greater Nicoya. (...) I would suggest before focusing on these 3 geographic areas, to mention the reasons why the next three sections are focused on those three particular geographic regions within the context of all Mesoamerica.

Before describing the aDNA studies in the Basin of Mexico, the Yucatán Peninsula and the Greater Nicoya, we have now explained that these geographical areas are the main focus of aDNA research. However, we also mention other areas of Mesoamerica where extensive archaeological and anthropological research have been done, but for which significant aDNA studies are still lacking. (e.g. the Gulf Coast and Oaxaca). Moreover, we describe how the North and the West remain understudied in relation to aDNA research (see lines 246-314). 

More info about the only individual with autosomal data would be very interesting, mainly, because all previous studies on aDNA and conclusions are based on mtDNA haplogroup frequencies considering few individuals that may not represent the ancient population. Thus, having autosomal data would be highly valuable.

Regarding this comment, we explain in the text (lines 801-808) that “The genomic data (..) revealed that they are genetically closer to present-day groups elsewhere in Central and South America rather than from the region near Belize. This result suggests that they formed part of a homogenous population associated with the Clovis culture that expanded southwards carrying the primary source of the ancestry of Central and South Americans”. Additionally, we already mentioned their mitochondrial haplogroups and that the two individuals from Saki Tzul are genetically related. The only piece of information left out relates to the Y chromosome haplogroups, which is of no relevance to the current work. 

Line 37: I am confused with the geographic reference. To what region do the authors refer when mentioning “northern North America” and “southern North America”?. Because in Scheib et al., 2018, the samples they sequence are from North America, but I would not consider that region northern North America. Do they mean Pacific and Atlantic Coast? I see that Moreno-Mayar used these terms (AB, NNA and SNA), but to mention dispersal routes. I suggest to better clarify these geographic regions.

The geographic references are described according to natural and political delimitations. The North American continent is the northern subcontinent of the Americas and it is composed of Canada, Greenland, the USA and Mexico, among other small countries and dependencies. The Arctic is composed of the northern portions of Alaska (USA), Northern Canada and Greenland. Therefore, when we mention northern North America, we are describing the northern portion of North America comprising Canada, the USA and Greenland outside the Arctic area. When we refer to southern North America, we are describing the southernmost part of North America that includes the southern border of the USA and Mexico. Central America refers to the area expanding from Guatemala and Belize to Panama, whereas the Caribbean includes the islands in the Caribbean Sea including those bordering the North Atlantic Ocean. Finally, the South American continent is the southern subcontinent of the Americas and it expands from Colombia to the Southern Cone as well as other dependencies. 

Line 86. I don’t understand what authors mean by: “and whose varied ancestry represented a unique source of genetic diversity in the continent.”

It is a way of describing that Mesoamericans probably had several civilisations with varied genetic backgrounds that were connected by commerce and farming, creating an invaluable source of genetic diversity that was partly lost during the Spanish conquest and the following colonial period.

Line 114: In my opinion the components of Native Mexicans should be better specified. In fact, the African component is minimal, and the European component has a much lower proportion than that found in cosmopolitan Mexicans. The differences in these three components should be stated or otherwise do not refer to Native Mexicans, but Mexicans in general.

Indeed, this description needed further explanation and the differences between the ancestry proportions in Mexicans have now been explained (see lines 207-210).

Line 146: you mention a work (ref 3) that analysed genomic data in Mesoamerica, but this reference earlier in the introduction was cited when talking on northern North America.

Posth et al. (2018) mainly focuses on South America, but also included three individuals from Belize (Central America) before the settling of Mesoamerica. This paper is referenced in the introduction following the sentence: “Within the Americas, the genetic history of the Arctic, northern North America and South America have been recently studied using aDNA”, as this work represents the biggest effort to date to study the genetic history of South America using nuclear data from aDNA. 

Figure 2. What do the red triangles mean?

The red triangles represent the volcano eruptions mentioned throughout the review. We have now included a description in the caption for clarity. 

Figure 3. What the different colors of the names of the sites represent?  Which are the difference between colored points with the black points?

The coloured points represent the archaeological sites included in each aDNA study. Black points represent archaeological sites where no aDNA data currently exists. For example, all the pink points mark the archaeological sites included in Ochoa-Lugo et al. (2016), whereas the yellow point indicates a different archaeological site (Hoyo Negro) from another study (Chatters et al. (2014). The black point on Tikal was included for its historical relevance, even though no aDNA study has so far been conducted. To make sure these points are clear, we have included a description in the caption.

Typos: Line 67: humans societies --> human societies; Line 156: Focused in --> Focused on

Both typos have been corrected.

Reviewer 2 Report

The paper is a nice review on the status of research regarding population genomics in Mesoamerica, specifically in light of results from ancient DNA.

This research is quite limited, and the authors point out that it is highly desirable to expand investigation in this understudied area, given its complex known history and the complex unknowns. The background provided on this is certainly very helpful for guiding future work.

I have only two small things: First, it would be good to emphasize a bit more the limitation of mtDNA, which most of the cited studies rely on. It remains a single locus among millions in the genome, and the sample sizes are often small, making it even more difficult to interpret the complex history of human populations. In other parts of the world, work on larger parts of the genome has shown a quite different picture to preliminary insights from mtDNA. This should make us cautious not to over-interpret the findings (which the author's don't do, so that is fine). Of course, the mtDNA is what is currently available, but the limitations should be clear. Second, a recent study proposed an early contact between central American and Oceanian populations (10.1038/s41586-020-2487-2), which may be worth discussing in this review.

Author Response

We would like to thank you for the constructive comments and valuable insights provided, which have contributed to strengthen this manuscript. We thoroughly edited the manuscript to correct spelling mistakes and to improve the readability of some sentences. We also addressed all of the points that you raised as detailed below.

First, it would be good to emphasize a bit more the limitation of mtDNA, which most of the cited studies rely on. It remains a single locus among millions in the genome, and the sample sizes are often small, making it even more difficult to interpret the complex history of human populations. In other parts of the world, work on larger parts of the genome has shown a quite different picture to preliminary insights from mtDNA. This should make us cautious not to over-interpret the findings (which the author's don't do, so that is fine). Of course, the mtDNA is what is currently available, but the limitations should be clear. 

We have now incorporated the following sentence  on the limitations of mtDNA research: “However, there are limitations associated with the study of mtDNA. Indeed, mtDNA represents the evolutionary history of the female population at a single locus and sample sizes are often small. Therefore, it is necessary to remain cautious when interpreting mtDNA results mainly when there is disagreement with those obtained from genome-wide analysis” (lines 232-236). 

Second, a recent study proposed an early contact between central American and Oceanian populations (10.1038/s41586-020-2487-2), which may be worth discussing in this review.

Following the reviewer’s suggestions, we have decided to mention the study performed by Ioannidis et al. (2020) by including the following sentence: “a recent study proposed a putative contact between Central Americans and Oceanian populations (AD ~1200) using modern DNA and some ancient individuals across the Americas (including the Saki Tzul individuals from the Maya Mountains)” (lines 925-928).

Round 2

Reviewer 1 Report

The authors addressed all my comments and suggestions and now the article is indeed more understandable and suitable for readers not specialised in that particular region. 

However, I still have two minor comments:

1) Now, since the sentence that referred to the first migration wave was removed (line 14 of the abstract). To not confuse the reader about the time of human presence in that area, I think that now it is necessary to specify: "The permanent settling" instead of just "The setlling", in the sentence of line 15.

2) The authors replied to my comment: Line 86. I don’t understand what authors mean by: “and whose varied ancestry represented a unique source of genetic diversity in the continent.” However, probably I was not clear enough in my comment. I did understand what the authors tried to say, but in my opinion the sentence should be clearer regarding ancestries/ genetic source... 

I propose to change the sentence: "and whose varied ancestry represented a unique source of genetic diversity in the continent".

and use the words used in their reply, which is much more clearer, I think:

"and whose varied genetic backgrounds created an invaluable source of genetic diversity in the continent."

Author Response

We addressed your two minor comments as detailed below.

1) I think that now it is necessary to specify: "The permanent settling" instead of just "The setlling", in the sentence of line 15.

This sentence in the abstract has been modified and now it says: “The permanent settling of Mesoamerica was accompanied by the development of agriculture and pottery manufacturing (2,500 BC–AD 150) …”.

2) I propose to change the sentence: "and whose varied ancestry represented a unique source of genetic diversity in the continent" and use the words used in their reply, which is much more clearer, I think: "and whose varied genetic backgrounds created an invaluable source of genetic diversity in the continent."

We have changed this sentence as you suggested and now it is as follows: “Mesoamerica was home to different human cultures connected by commerce and farming, and whose varied genetic backgrounds created an invaluable source of genetic diversity in the continent”.